# A Novel Screen for Expression Regulators of the Telomeric Protein TRF2 Identified Small Molecules That Impair TRF2 Dependent Immunosuppression and Tumor Growth

**DOI:** 10.3390/cancers13122998

**Published:** 2021-06-15

**Authors:** Mounir El Maï, Serena Janho dit Hreich, Cedric Gaggioli, Armelle Roisin, Nicole Wagner, Jing Ye, Pierre Jalinot, Julien Cherfils-Vicini, Eric Gilson

**Affiliations:** 1Institut National de la Santé et de la Recherche Médicale (INSERM) U1081, Université Côte d’Azur, Centre National de la Recherche Scientifique (CNRS) UMR7284, Institute for Research on Cancer and Aging, Nice (IRCAN), 06107 Nice, France; Mounir.El-Mai@unice.fr (M.E.M.); serena.janho-dit-hreich@etu.univ-cotedazur.fr (S.J.d.H.); Cedric.Gaggioli@unice.fr (C.G.); 2Laboratory of Biology and Modelling of the Cell (LBMC), ENS de Lyon, Univ Lyon, INSERM U1210, CNRS UMR 5239, Université Claude Bernard Lyon 1, 46 Allée d’Italie Site Jacques Monod, 69007 Lyon, France; armelle.roisin@ens-lyon.fr (A.R.); pierre.jalinot@ens-lyon.fr (P.J.); 3Institut National de la Santé et de la Recherche Médicale (INSERM), Université Côte d’Azur, Centre National de la Recherche Scientifique (CNRS), Institute of Biology Valrose, 06108 Nice, France; Nicole.Wagner@unice.fr; 4International Research Project “Hematology, Cancer and Aging”, Póle Sino-Français de Recherche en Sciences du Vivant et Génomique, Shanghai Ruijin Hospital, Shanghai Jiao Tong University School of Medicine, Shanghai 200025, China; yj11254@rjh.com.cn; 5Department of Medical Genetics, Archet 2 Hospital, FHU Oncoage, CHU of Nice, 06000 Nice, France

**Keywords:** TRF2, cancer, aging, cell-based screening assay, neo-angiogenesis, immune suppression

## Abstract

**Simple Summary:**

The telomeric protein TRF2 (Telomeric repeat-binding factor 2) is upregulated in human cancers and associated with poor prognosis. TRF2 oncogenic properties rely on its intrinsic telomere protective role, but also on cell extrinsic effects through immunosuppressive and angiogenic activities. Therefore, targeting TRF2 appears as a promising therapeutic anti-cancer strategy. In this study, we developed a cell-based method to screen for TRF2 inhibitors allowing us to identify two compounds that blunt the TRF2 pro-oncogenic properties in vivo.

**Abstract:**

Telomeric repeat-binding factor 2 (TRF2) is a subunit of the shelterin protein complex, which binds to and protects telomeres from unwanted DNA damage response (DDR) activation. TRF2 expression plays a pivotal role in aging and cancer, being downregulated during cellular senescence and overexpressed during oncogenesis. Cancers overexpressing TRF2 often exhibit a poor prognosis. In cancer cells, TRF2 plays multiple functions, including telomere protection and non-cell autonomous roles, promoting neo-angiogenesis and immunosuppression. We present here an original screening strategy, which enables identification of small molecules that decrease or increase TRF2 expression. By screening a small library of Food and Drug Agency (FDA)-approved drugs, we identified two molecules (AR-A014418 and alexidine·2HCl) that impaired tumor growth, neo-angiogenesis and immunosuppression by downregulating TRF2 expression in a mouse xenograft model. These results support the chemotherapeutic strategy of downregulating TRF2 expression to treat aggressive human tumors and validate this cell-based assay capable of screening for potential anti-cancer and anti-aging molecules by modulating TRF2 expression levels.

## 1. Introduction

Telomeres are specialized nucleoprotein structures found at the ends of linear chromosomes that are regulated by telomere-associated factors such as telomerase, shelterin protein complexes and non-coding telomeric repeat-containing RNA [1]. When properly regulated, telomeres protect chromosomes against instability and senescence. Telomeric DNA shortening occurs as part of programmed physiological development and aging [2]. However, excessive telomere DNA shortening drives rare progeroid syndromes, such as dyskeratosis congenita [3]. Moreover, dysregulated telomere states are implicated in numerous diseases that are common throughout the general population, including almost all types of cancer and several degenerative diseases [4]. Thus, developing pharmacological treatments to target specific telomere components is promising to prevent and treat such diseases.

As far as telomere and cancer are concerned, vital DNA damage response (DDR) checkpoints sometimes fail; this can lead to excessive telomeric DNA shortening and aberrant chromosome rearrangements, which in turn can contribute to oncogenesis. Furthermore, upregulation of telomerase is a key event in the development of about 90% of all cancers, as this can confer unlimited growth to cancer cells [5]. For this reason, telomerase inhibition has been the target of several studies seeking to develop cancer treatments [6]. Despite recent progress, there are some limitations to the clinical use of anti-telomerase drugs. For example, to halt cancer cell proliferation, a critically short telomeric DNA length must be reached, and the anti-oncogenic effects of shortened telomeres are lost in the absence of the *p53* tumor suppressor gene [7,8]. This lag period reduces therapeutic efficacy and increases pro-aging side effects by limiting cell renewal and favoring the activation of alternative recombination-based telomere elongation mechanisms [9,10]. Therefore, anti-telomerase strategies may be better suited for targeting cancer cells that already harbor critically short telomeres [11].

In addition to telomerase, changes in the expression and activities of shelterin complex subunits (TRF1, TRF2, RAP1, TIN2, TPP1 and POT1) are involved in tumorigenesis, and in some cases independently of telomere length [12,13,14,15,16]. Thus, targeting shelterin for cancer treatment could be an interesting alternative to anti-telomerase interventions. The shelterin subunit TRF2 represents an interesting candidate; TRF2 is overexpressed in several human malignancies, both in cancer and vascular cells and this is typically associated with a poor prognosis [17,18,19,20]. Notably, TRF2 overexpression can promote tumorigenesis non-cell autonomously, leading to immunosuppression and neo-angiogenesis [13,18,19,20]. In particular, TRF2 upregulation creates a potent immunosuppressive microenvironment. By altering the expression of heparan sulfate proteoglycans, TRF2 directly recruits and activates myeloid-derived suppressive cells (MDSCs) through TLR2 pathway [21]. While TRF2 overexpression in cancer cells strongly inhibits NK cell recruitment in the tumor microenvironment by the regulation of HSPG synthesis [13], the TLR2 activation of MDSC induced by the overexpression of TRF2 leads to a powerful inhibition of the NK cell immunosurveillance with a strong decrease of NK cells degranulation, IFNgamma production and killing [21]. Thus, TRF2 overexpression strongly blunt early stage of anti-tumor immunosurveillance by directly inhibiting NK cells recruitment and indirectly the functionality of NK cells through the shaping of a MDSC dependent immunosuppressive microenvironment. Consequently, targeting TRF2 in cancers could be a valuable multi-hit strategy via cell-autonomous and non-cell-autonomous processes by promoting senescence as well as impairing neo-angiogenesis and immune escape.

To date, all identified small compounds that target TRF2 impair its ability to protect chromosome ends from DDR thus with potential pro-aging side effects [22]. Our previous work demonstrated that a partial reduction of TRF2 expression can reverse tumorigenicity in mouse models through non-cell-autonomous effects, without inducing DDR [13]. We therefore reasoned that focusing on reducing excess TRF2 in cancers resulting from its overexpression could be an interesting strategy to treat cancer without pro-aging side effects. In this study, we present an original screening platform to identify small compounds targeting TRF2 stability. We show that two top hits inhibited the tumorigenic activity of TRF2 overexpression. This study demonstrated that TRF2 expression can be pharmacologically modulated, and that our screening assay is a reliable method for the selection of drugs capable of modulating TRF2 expression levels, with potential anti-cancer and anti-aging properties.

## 2. Materials and Methods

### 2.1. Cells

Human embryonic kidney (HEK) 293-T cells (ATCC CRL-1573) and BJ-HELTRas cells [13] were grown in Dulbecco’s Modified Eagle’s Medium (DMEM) (Lonza, Levallois-Perret, France) supplemented with 10% fetal calf serum (FCS), 100 IU/mL penicillin and 100 µg/mL streptomycin (Invitrogen, Cergy Pontoise, France).

### 2.2. SDS-PAGE and Western Blotting

Total cell lysates were prepared, separated by electrophoresis and blotted as previously described [14]. Briefly, cells were harvested and lysed in lysis buffer (8.76 g/L NaCl, 10 mM Tris-HCL pH 7.2, 0.1% SDS, 0.1% Triton X-100, 10 g/L sodium deoxycholate, 5 mM ethylenediaminetetraacetic acid (EDTA), 10 µg/mL leupeptine, 1 mM AEBSF and 19 µg/mL aprotinin). Samples were then titrated using BCA protein assay kit (Interchim, Monluçon, France). Samples (60 µg/lane) were heated at 95 °C for 5 min in loading buffer (500 mM Tris-HCl, 100 mM DTT, 2% SDS, 0.1% bromophenol blue, 10% glycerol, pH 6.8). Then samples were loaded on 10% polyacrylamide gels and run for 45 min at 160 V. Proteins were then transferred onto Immobilon-FL membranes (Millipore, Upstate, New York, USA) using Trans-Blot SD Semi-Dry Electrophoretic Transfer Cell (Bio-Rad, Hercules, CA, USA). Membranes were blocked for 1 h in Intercept Blocking buffer (LI-COR, Lincoln, NE, USA) prior to overnight incubation at 4 °C with a mix of primary antibodies (mouse IgG1 anti-TRF2 diluted at 1:250; and rabbit IgG anti-Beta-actin diluted at 1:10,000) in Intercept Blocking buffer containing 0.5% Tween20. After being washed in PBS, 0.1%Tween20, membranes were incubated for 1 h at room temperature in Intercept blocking buffer containing 0.25% Tween20 with a mix of secondary antibodies: goat-anti-mouse IRDye 680 for anti-TRF2 antibody and goat anti-rabbit IRDye 800CW for anti-Beta-actin antibody (1:15,000 dilution). Primary and IRDye secondary antibodies that were used are listed below in the antibody table (Table 1). Finally, protein bands were visualized using LiCor Odyssey 9120 imaging system (LI-COR). TRF2 expression was measured by normalizing TRF2 band intensity to the intensities of the actin band and the background.

### 2.3. Cloning Strategy

The human TRF2 cDNA sequence was cloned between the BamHI/BclI restriction sites of a lentiviral SFFV-GPR plasmid that is a HIV-SFFV-GFP-WPRE derivative [23]. The SFFV-GPR plasmid contains a spleen focus forming virus (SFFV) promoter controlling a GPR polycistronic gene comprising cDNA sequences for green fluorescent protein (GFP), a puromycin resistance protein and Tag-red fluorescent protein (RFP)-T (S158T mutated Tag-RFP), each separated by E2 and T2 *Picornaviridae* sequences. hTRF2 cDNA was inserted at the C-terminal region of RFP-T, to enable the expression of an RFP-TRF2 fusion protein.

### 2.4. Lentivirus Production

For lentivirus production, 5 × 10^6^ HEK 293-T cells were transfected with 8.6 µg of empty SFFV-GPR or RFP-TRF2-expressing SFFV-GPR vector, 8.6 µg of Lenti-Delta 8.91 and 2.8 μg of VSV-g via calcium phosphate-mediated transfection. Transfected cells were cultured in DMEM supplemented with 10% FCS at 37 °C with 5% CO_2_ in 10-cm dishes. Supernatants containing the newly constructed viruses were collected after 48 h, then passed through a 0.45-μm Millipore filter. After virus titration, a 1:1 (virus:cell) ratio was used to infect the BJ-HELTRas cell line, chosen for its resistance to TRF2 defects [13]. A clone that expressed GFP and the fused RFP-TRF2 protein to moderate levels was then isolated by fluorescence-activated cell sorting (FACS).

### 2.5. Flow Cytometry Screening

BJ-HELTRas clonal line cells containing the SFFV-GPR vector and expressing the RFP-TRF2 fusion protein were cultured in 96-well plates in DMEM medium supplemented with 10% FCS and 1% penicillin/streptomycin. After 24 h of drug treatment, cells were then trypsinized and washed in phosphate buffered saline (PBS) containing 0.5 mM EDTA and 2% FCS before fixing with 0.5% formaldehyde (FA). Fixed cells were then subjected to flow cytometry using a FacsCalibur high-throughput sampler (BD Biosciences, Franklin Lakes, NJ, USA).

### 2.6. Real-Time Quantitative Polymerase Chain Reaction (RT-qPCR)

Total RNA was isolated using RNeasy Mini Kit (Qiagen, Venlo, Netherlands). Reverse transcription was performed using Superscript II reverse transcriptase (Invitrogen) with 1 μg of total RNA. The expression of each gene was normalized to that of GAPDH. The following primers were used: hTRF2 Fw 5′-GCTGCCTGAACTTGAAACAGT-3′; hTRF2 Rv 5′-CCGTTCTCAACCAACCCCTC-3′; hGAPDH Fw 5′-AGCCACATCGCTCAGACAC-3′; hGAPDH Rv 5′-GCCCAATACGACCAAATCC-3′.

### 2.7. AlamarBlue

In a 96-well flat-bottom plate, 5 × 10^3^ cells per well were seeded in 200 μL DMEM supplemented with 10% FCS. At 24 h after drug treatment, 10 µL of AlamarBlue (Bio-Rad) was added and absorbance was measured at 570 nm and 600 nm for 36 h in a Spectrostar Nano plate reader (BMG Labtech, Ortenberg, Germany). The oxidation–reduction percentage of AlamarBlue was determined as described by the manufacturer.

### 2.8. Animals

Experiments were performed on 8- to 12-week-old NMRI nude female mice from Janvier Labs (France). All mouse experiments were conducted according to local and international institutional guidelines and were approved by either the Animal Care Committee of the IRCAN and the regional (CIEPAL Cote d’Azur #187 and #188) and national (French Ministry of Research #03482.01/02482.2 and # 02973.01/02973.2) authorities.

### 2.9. Tumor Growth Experiments

Each NMRI nude mouse was injected subcutaneously in the back with 1 × 10^6^ BJ-HELTRas cells suspended in 100 µL of PBS (*n* = 8 mice per group). The mice were treated on days 16, 18, 20 and 22 with intraperitoneal injections of 100 µL of DMSO (45%), alexidine·2HCl (1 mg/kg) or AR-A014418 (5 mg/kg), then followed up until day 26. The tumor appearance was assessed by palpation every day. Tumor size was measured every 2–3 days using a caliper. Tumor volume was then determined using the hemi-ellipsoid formula: π × (L × l × h)/6, where L corresponds to the length, l to the width and h to the height of the tumor, respectively.

### 2.10. Matrigel Plug Assays

BJ-HELTRas cells were treated with DMSO (1%), alexidine·2HCl (1 µM) or AR-A014418 (10 µM) for 2 days. Then, 100 μL of 1 × 10^6^ treated cells suspended in PBS together with 400 μL of growth-factor-reduced Matrigel (Corning, New York, USA) were inoculated subcutaneously into the back of NMRI nude mice under isoflurane anesthesia. At day 5 post-inoculation, the Matrigel plugs were harvested, and infiltrating cells were collected by enzymatic dissociation via dispase (Corning), collagenase A (Roche, Bâle, Switzerland) and DNAse I (Roche) digestion for 30 min at 37 °C [13]. Cells were saturated for 15 min on ice with Fc-Block anti-CD16/CD32 antibodies (clone 2.4G2) prior to staining with coupled antibodies for 30 min at 4 °C. The conjugated antibodies used are listed in the antibody table. Cells were washed in PBS with 0.5 mM EDTA, 2% FCS and fixed with 0.5% FA. Stained cells were analyzed using an ARIA III cytometer with DIVA6 software (BD Biosciences) and FlowJo 10 (LLC).

### 2.11. Statistics

All graphs and statistical analyses were produced using GraphPad Prism software (San Diego, CA, USA). All results are represented as the mean ± standard deviation (s.d.) or mean ± standard error of the mean (SEM). Significant differences between the means were determined using the Mann–Whitney two-tailed test. The log-rank (Mantel–Cox) test was used to determine tumor take. For each test, *p* < 0.05 was considered statistically significant.

## 3. Results

### 3.1. Identification of TRF2 Inhibitory Molecules

To screen for drugs capable of modulating TRF2 protein levels, we used a lentiviral system to co-transcribe the GFP gene and RFP fused to the N-terminus of TRF2. Following a general strategy described elsewhere [23], we constructed a GRT lentivirus, in which the SFFV promoter controlled the expression of a poly-cistronic gene encoding GFP, a puromycin resistance protein and either RFP-TRF2 or RFP only as a control (Figure 1A). Consequently, all transduced cells expressed both GFP and RFP-TRF2. This system was designed to identify drugs that modulate TRF2 expression, by measuring RFP intensity using flow cytometry, while GFP intensity serves as an internal control for the transcription of the reporter construct.

We transduced the GRT lentiviruses into human BJ-HELTRas fibroblasts that were immortalized by SV40 and hTERT and rendered oncogenic by Ras v12 [13]. To facilitate measurements of both up- and down-regulation of TRF2, a puromycin-resistant clone that moderately expressed both GFP and RFP-TRF2 proteins was isolated using FACS sorting and named GRT-BJ-HELTRas (Figure 1A). As a positive control, we treated GRT-BJ-HELTRas cells with 10 µM gemcitabine, a previously described modulator of TRF2 stability [24], for 24 h. While no RFP modulation was detected in cells transduced with empty vector, a significant reduction in the RFP/GFP mean fluorescence intensity (MFI) ratio was observed in gemcitabine-treated GRT-BJ-HELTRas cells compared to the DMSO-treated control (82% of the control; *p* < 0.0001) (Figure 1B). Moreover, gemcitabine specifically diminished RFP-TRF2 protein levels without affecting GFP levels (Appendix A). Of note, we observed here that gemcitabine is reducing TRF2 level in contrast to the published work [24], a difference that may be explained by cell type differences in the DNA damage response induced by gemcitabine since TRF2 stability can be altered in a p53-dependent manner [25]. This effect was further confirmed by Western blotting analyses where we observed that endogenous TRF2 levels were affected by gemcitabine treatment on untransduced BJ-HELTRas cells (Appendix A). Therefore, GRT-BJ-HELTRas cells enabled detection of specific variations in TRF2 protein levels induced by drug treatment.

Using flow cytometry, we then screened 396 Food and Drug Administration (FDA)-approved pharmacological compounds capable of targeting six main categories of biological processes: ion channels, phosphatases, kinases, epigenetic factors, nuclear receptor ligands and the Wnt pathway (Appendix A). GRT-BJ-HELTRas cells were first treated with 10 µM of each of the 396 compounds or DMSO for 24 h prior to flow cytometry analysis (Appendix A). In this case, 84 compounds were found to modulate TRF2 protein levels (Appendix A, right panel; Appendix A), and were used for a secondary screen (Appendix A; Appendix A). In this secondary screen, in addition to treating GRT-BJ-HELTRas cells with 10 µM of the selected compounds, non-transduced BJ-HELTRas cells or BJ-HELTRas cells transduced with an empty vector were treated similarly to discard false positives emitting red or green autofluorescence or affecting RFP or GFP proteins. The 18 best compounds were then selected to determine their ability to modulate the expression of endogenous TRF2 in non-transduced BJ-HELTRas cells, based on Western blotting (Appendix A, Appendix A). We defined hits as compounds modulating by at least 20% of TRF2 dosage (either up or down). Using these criteria, from the 18 drugs evaluated by Western blotting, 9 of them reduced and one increased endogenous TRF2 protein levels (Appendix A, Appendix A). We then decided to choose two compounds among these drugs for in vivo experiments to determine whether they could counteract the pro-oncogenic effects of TRF2 overexpression. To avoid DDR activation and side effects in non-tumorigenic cells, we selected compounds that neither reduced to the highest nor to the lowest levels TRF2 dosage. Among them, AR and AD were corresponding to those criteria. Both compounds downregulated RFP-TRF2 in GRT-BJ-HELTRas cells as analyzed by flow cytometry (Figure 1C), endogenous TRF2 protein levels as shown by Western blotting analysis (Figure 1D; Appendix A) and *TERF2* mRNA levels as determined via RT-qPCR (Figure 1E). Moreover, treatment with the respective LD50 concentrations of AR and AD reduced *TERF2* mRNA expression in both BJ-HELTRas cells and in TRF2-overexpressing BJ-HELTRas cells (Appendix A–C).

### 3.2. AR-A014418 and Alexidine·2HCl Reversed the Tumorigenicity Conferred by High TRF2 Expression Levels

We next evaluated the impact of AR and AD on tumor growth. To control for their ability to target TRF2-overexpressing cancers, we analyzed the potential anti-tumorigenic effects of AR and AD on both standard BJ-HELTRas cells and TRF2-overexpressing BJ-HELTRas cells. BJ-HELTRas cells were transduced with TRF2 lentiviral vector or empty vector, then injected subcutaneously into nude mice. The mice were then treated with DMSO, 1 mg/kg AD or 5 mg/kg AR at days 16, 18, 20 and 22 post-injection [26,27] (Figure 2A). As previously reported [13,21], TRF2 overexpression promoted tumor initiation and growth in vivo (Figure 2B–D; *p* < 0.05). Treatment with neither AR nor AD impacted tumor volume in BJ-HELTRas cells transduced with the empty vector (Figure 2E, upper panels) neither tumor growth rate (Figure 2F–H). By contrast, both drugs induced a significant decrease in the tumor volume of TRF2-overexpressing xenografted tumors, leading to significantly smaller tumor sizes at day 26 (Figure 2E, middle panels; *p* < 0.0001) but also of the tumor growth rate (Figure 2F–H) at each time-point. Furthermore, the volume and growth rates of TRF2-overexpressing tumors treated by the two drugs returned to levels similar to those of the control tumors, thus demonstrating the TRF2-specific selectivity of AR and AD (Figure 2E, lower panels). These results show that AR and AD treatment reversed specifically the tumorigenicity conferred by TRF2 overexpression.

### 3.3. AR-A014418 and Alexidine·2HCl Counteracted TRF2-Dependent Immunosuppression and Neo-Angiogenesis

To determine whether the antitumor effects of AR and AD could target the non-cell autonomous oncogenic properties of TRF2, we examined immune cell infiltration and activation as well as angiogenesis in treated tumors. We treated standard and TRF2-overexpressing BJ-HELTRas cells (Appendix A) with 1 µM of AD, 10 µM of AR or DMSO for 48 h before their subcutaneous injection with Matrigel into nude mice. Matrigel plugs were then collected at 5 days post-injection to analyze the tumor microenvironment by flow cytometry (Figure 3A). As expected [13,21], TRF2 overexpression did not change global immune cell (CD45+ cell) infiltration (Figure 3B; Appendix A), but did inhibit natural killer (NK) cell recruitment (Figure 3C; Appendix A) and NK functionality (Figure 3C–E; Appendix A), and increase MDSC infiltration (Figure 3F). In TRF2-overexpressing BJ-HELTRas cells specifically, treatment with either drug increased global immune infiltration (Figure 3B; Appendix A), as well as the quantity and functionality of intra-tumoral NK cells (Figure 3C–E; Appendix A). Notably, both drugs fully rescued the inhibition of NK cell-mediated immune surveillance (CD107a+ and CD69+ NK cells) induced by TRF2 overexpression (Figure 3C). This was associated with a dramatic decrease in MDSC recruitment (Figure 3F; Appendix A) and with increased recruitment of monocytes and macrophages (Figure 3G).

As previously reported [14,20], tumor angiogenesis was higher in TRF2-overexpressing tumors compared to control tumors. While TRF2 overexpression increased the quantity of CD31+ CD45− endothelial cells within the tumor bed (Figure 3H; Appendix A), no difference was detected between empty vector or TRF2-overexpressing tumors after treatment with either drug.

## 4. Discussion

Here, we report the development of a cell-based assay to screen for compounds that modulate expression of the telomeric protein TRF2. By screening a small library of FDA-approved molecules, we identified compounds that could either increase or decrease TRF2 expression levels. We discovered that AD and AR downregulated TRF2 and exhibited anti-tumorigenic activity specific for tumor cells overexpressing TRF2. Further demonstrating their TRF2-specific activity, AR and AD treatment rescued immunosuppression and neo-angiogenesis conferred by TRF2 overexpression. Strikingly, the fact that the global immune infiltration is increased for TRF2 overexpressing tumors after AR or AD treatment suggest that those drugs are more potent to enhance immune response when TRF2 is overexpressed. Those drugs inhibit the immunosuppressive effect of TRF2 overexpression by restoring NK cell functionality (CD107a and CD69+ NK cells) and strongly decrease MDSC infiltration. This suggest that those drugs blunt the TRF2 dependent specific program that trigger immune escape and immunosuppression and may enhance the release of Danger Associated Molecules (DAMP) that enhance immune response specifically when TRF2 is overexpressed. Of note, we observe that AR and AD rescued the immunosuppressive and pro-angiogenic functions of TRF2 overexpression, two characteristics of TRF2 overexpression that we previously showed as DDR independent. Thus, we hypothesized that the AR and AD effects on tumor growth are DDR independent, a mechanism that remain to be fully described in further studies. The present proof-of-concept study provides evidence that pharmacological reduction of TRF2 expression could be a valuable anti-cancer strategy. 

Even though the screening method considered TRF2 protein levels independently of transcript levels, both drugs decreased endogenous *TERF2* mRNA levels, implying that AR and AD target multiple levels of TRF2 regulation. Supporting this, AR is an inhibitor of Wnt signaling [28], which is an activator of *TERF2* transcription [29]. More generally, the drugs targeting the Wnt signaling pathway were enriched during the screening steps (Appendix A–D). How AD, a mitochondria-targeting agent, affects TRF2 expression remains to be determined. Since cancers that exhibit high TRF2 levels have a poor prognosis and exhibit increased resistance to chemotherapy [21], AR and AD are therapeutic agents of interest for such tumors. The potential to uncouple the telomeric and pro-oncogenic activities of TRF2 [13] raises the possibility of pharmacologically downregulating TRF2, thus conferring multi-hit anti-cancer benefits without deleterious pro-aging side effects. Therefore, future studies are warranted to determine the synergistic effects of these drugs on TRF2 expression levels in clinical studies.

Even though TRF2 is upregulated in various human cancers [13,21], its expression is downregulated during both normal and pathological aging of many tissues [30,31]. Moreover, several reports involving mouse models of TRF2 dysregulation emphasize the importance of TRF2 at the crossroads between aging and cancer [31,32,33,34,35]. Therefore, the molecules identified by the screening procedure described here are interesting drug candidates, both as anti-cancer agents for TRF2 downregulation as confirmed in this study, and potentially as anti-aging agents by upregulating TRF2.

## Figures and Tables

**Figure 1 cancers-13-02998-f001:**
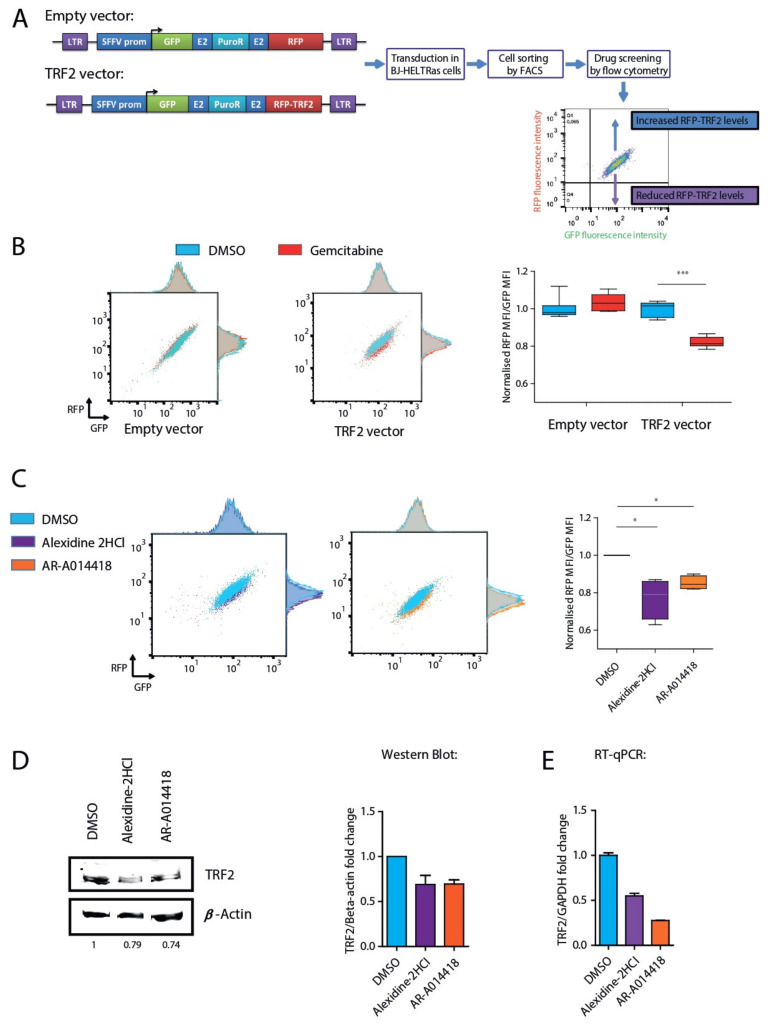
Identification of drugs that affect TRF2 expression using fluorescence-based drug screening. (**A**) Schematic representation of the drug screening experimental design. Lentiviral constructs that contained a poly-cistronic gene encoding green fluorescent protein (GFP) and either red fluorescent protein (RFP) alone (empty vector) or a fusion of RFP protein at the N-terminal region of TRF2 (TRF2 vector) were transduced into the BJ-HELTRas cell line. Clones sorted using fluorescence-activated cell sorting (FACS) were then used to screen for drugs that modified TRF2 protein levels. (**B**) Proof of concept using the TRF2-regulating drug gemcitabine as a positive control. Representative RFP/GFP flow cytometry density plots of empty vector- or TRF2 vector-transduced BJ-HELTRas cells treated with either DMSO or gemcitabine (10 µM) (left panel). Box-plot quantification of RFP (empty vector) or RFP-TRF2 fusion (TRF2 vector) protein after treatment with DMSO or gemcitabine (right panel) (*n* = 7; *** *p* < 0.001; two-tailed Student’s *t* test). (**C**) Representative RFP/GFP flow cytometry plots of TRF2 vector-transduced BJ-HELTRas cells treated with 10 µM of alexidine·2HCl, AR-A014418 or DMSO (left panel). Box-plot quantification of RFP-TRF2 fusion (TRF2 vector) protein levels following treatment with DMSO, alexidine·2HCl or AR-A014418 (right panel) (*n* = 4; * *p* < 0.05; Mann-Whitney test). (**D**) Representative Western blotting showing reduced TRF2 protein levels following treatment with 10 µM alexidine·2HCl or AR-A014418 compared to DMSO control treatment in untransduced BJ-HELTRas cells (left panel; full Western blotting image is presented in Appendix A). Quantification of TRF2 protein levels after treatment with 10 µM alexidine·2HCl or AR-A014418 compared to DMSO control treatment (right panel) (*n* = 2; mean + standard error of the mean). (**E**) Quantification of TRF2 mRNA levels analyzed by RT-qPCR after treatment with 10 µM alexidine·2HCl or AR-A014418 compared to DMSO control treatment.

**Figure 2 cancers-13-02998-f002:**
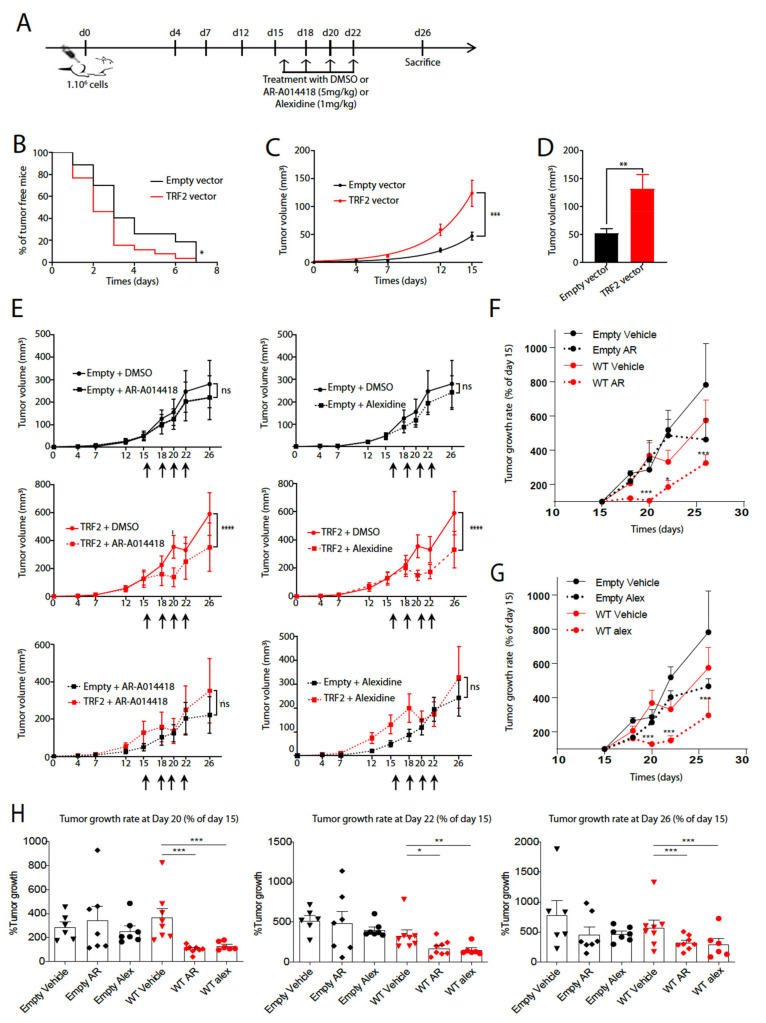
In vivo alexidine·2HCl or AR-A014418 treatment inhibited TRF2-dependent tumorigenesis. (**A**) Schematic representation of the experimental design. BJ-HELTRas cells either overexpressing TRF2 (TRF2 vector) or not (empty vector) were injected subcutaneously (1 × 10^6^ cells/100 µL) into NMRI nude mice. The mice were then treated with DMSO, alexidine·2HCl (1 mg/kg) or AR-A014418 (5 mg/kg) at days 16, 18, 20 and 22 post-injection, then followed up until day 26 (*n* = 8 mice per group). (**B**–**D**) The percentage of tumor-free mice (**B**) and tumor volumes (**C**) were determined at the indicated time points in mice injected with BJ-HELTRas cells overexpressing TRF2 (TRF2 vector) or empty vector. Tumor take based on palpability was determined at the indicated time-points and is represented as a percentage of tumor-free mice. *p* values were determined using the log-rank Mantel-Cox test (* *p* < 0.05). Tumor volumes were assessed at different time-points (mean ± standard deviation); tumor volume at day 15 is represented in (**D**). *p* Values were determined using the Mann–Whitney test (* *p* < 0.05; ** *p* < 0.005; *** *p* < 0.001). (**E**) Tumor volumes were followed as in (**C**) after repetitive treatments with DMSO, alexidine·2HCl (1 mg/kg) or AR-A014418 (5 mg/kg). *p* Values were determined using the Mann–Whitney test (**** *p* < 0.0001; ns: not significant). (**F**–**H**) Tumor growth rate of AR-A014418 (5 mg/kg) treated mice (**F**,**H**) or alexidine·2HCl (1 mg/kg) (**G**,**H**) treated mice were determine by considering the last day before treatment for empty vector or TRF2 vector as 100%. Variation of the growth rate in both treatment over the time are represented in (**F**,**G**) and for each time-point (**H**). *p* Values were determined using the Mann–Whitney test (* *p* < 0.05; ** *p* < 0.005; *** *p* < 0.001).

**Figure 3 cancers-13-02998-f003:**
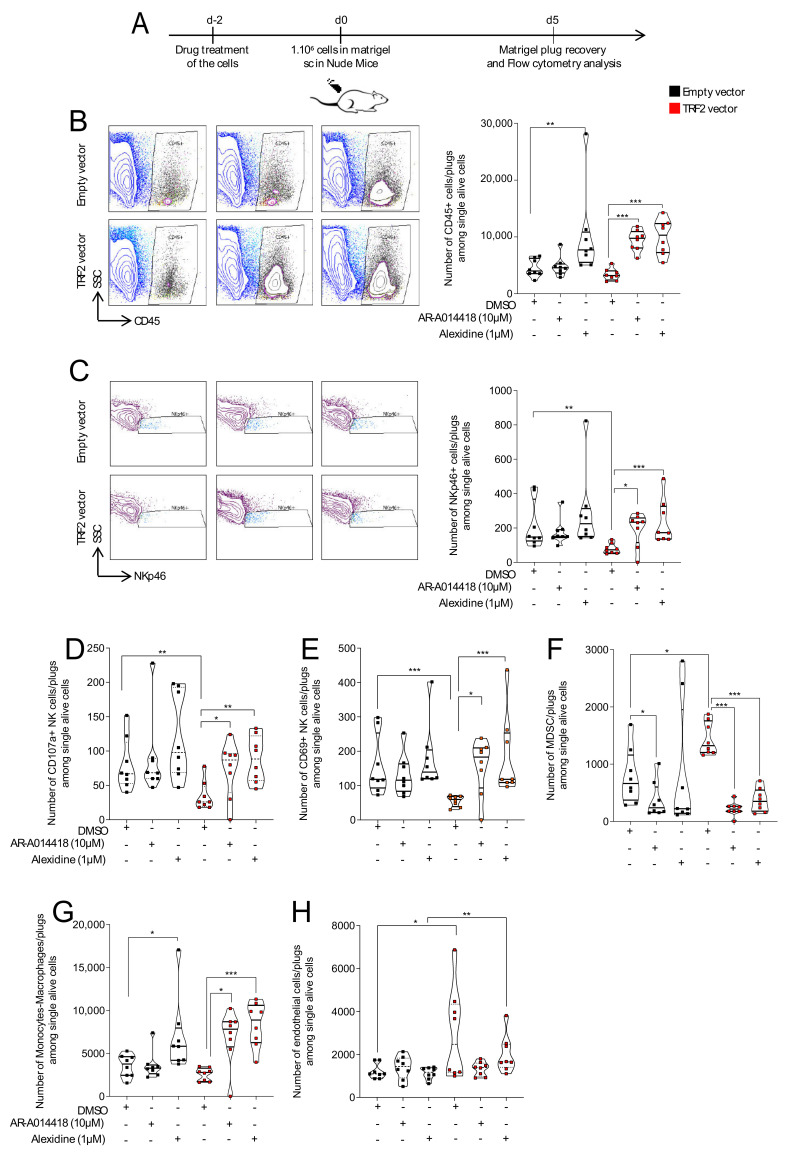
Treatment with alexidine·2HCl or AR-A014418 reverted the TRF2-mediated immunosuppressive microenvironment and restored immune cell infiltration. (**A**) Schematic representation of the experimental design. BJ-HELTRas cells overexpressing TRF2 (TRF2 vector) or empty vector were treated with 1 µM of alexidine·2HCl or 10 µM of AR-A014418 or DMSO at 2 days before subcutaneous injection with Matrigel (1 × 10^6^ cells) into NMRI nude mice (*n* = 8). Immune and endothelial cell infiltration was then evaluated at 5 days post-injection by flow cytometry. (**B**–**H**). Flow cytometry analysis of the immune infiltration of the Matrigel plug. The numbers of immune cells infiltrating the Matrigel plugs among live cells are shown. Total immune cell infiltration (CD45+ cells) is shown in (**B**); natural killer (NK) cell (NKp46+ cells) infiltration is shown in (**C**); activated NK cell (CD107a+ and CD69+ NK cells) infiltration is shown in (**D**,**E**); myeloid-derived suppressor cell (MDSC; CD11b+ GR1+)) infiltration is shown in (**F**); monocyte-macrophage infiltration is shown in (**G**) and endothelial cell infiltration is shown in (**H**). *p* Values were determined using the Mann-Whitney test (* *p* < 0.05, ** *p* < 0.005, *** *p* < 0.001; *n* = 8 mice per group).

**Table 1 cancers-13-02998-t001:** Antibodies.

Specificity	Company	Clone	Species	Isotype	Fluorochrome	Reference
anti CD107a FITC	BD Biosciences	1D4B	Rat	IgG2a/k	FITC	553793
anti CD11b	BD Biosciences	M1/70	Rat	IgG2b	APC-H7	550993
Anti CD69 PE-Cy7	BD Biosciences	H1.2F3	Hamster	IgG1/K	PE-Cy7	552879
Anti NKp46 CD335	BD Biosciences	29A1.4	Rat	IgG2a	Alexa 647	560755
Anti-CD3e	Biolegend	145-C11	Armenin Hamster	IgG	PerCP	100302
Anti-CD45	BD Biosciences	30-F11	Rat	IgG2b, κ	AF700	560510
Anti-F4/80	Biolegend	BM8	Rat	IgG2a, κ	BV510	123135
Anti-Ly6C/Ly6G (Gr-1)	BD Biosciences	RB6-8C5	Rat	IgG2b, κ	PE	553128
anti-CD31	BD Biosciences	MEC 13.3	Rat	IgG2a, κ	BV421	562939
TRF2	Imgenex	4A794.15	Mouse	IgG1, κ	N.A.	IMG-124A
Beta-Actin	Abcam		Rabbit	IgG	N.A.	Ab8227
Anti-mouse	Licor		Goat	IgG	IRDye 680	926-32220
Anti-Rabbit	Licor		Goat	IgG	IRDye 800CW	926-32211

## Data Availability

Further information and requests for resources and reagents should be directed to and will be ful-filled by the lead contact, E.G. (Eric.GILSON@unice.fr).

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
