# Peer review of "A Novel Screen for Expression Regulators of the Telomeric Protein TRF2 Identified Small Molecules That Impair TRF2 Dependent Immunosuppression and Tumor Growth"

_cancers, 2021, doi:10.3390/cancers13122998_

Round 1

Reviewer 1 Report

In this work, Dr. Mai and colleagues aimed to identify compounds to counteract TRF2 oncogenic effects. The topic is quite interesting as TRF2 has been associated with poor outcomes and immunosuppression in several cancers.

Authors stemmed from their previous findings to devise a strategy for protein reduction instead of inhibition, thus avoiding DDR and pro-aging side effects, which is an interesting and original approach.

The model was BJ-HELTRas centered (immortalized oncogenic fibroblasts). Overall, the manuscript is well written and data well presented. Methods and controls were adequate.

However, despite authors previously shown that TRF2 expression can reverse tumorigenicity in mouse models through non-cell-autonomous effects, without inducing DDR (doi:10.1038/ncb2774,) a major concern is that although this study demonstrates a possible anti-oncogenic role of AR and AD by decreasing TRF2, the authors did not demonstrate that TRF2 reduction by AR and AD does not have DDR effects. Analysis of DDR proteins in GRT-BJ-HELTRas cels under AR and AD, for example, should be included.

Additionally, there are some aspects that, if addressed, may improve the quality of the manuscript prior to consideration for publication.

L246, SupFig 1E: It is not clear what were, if any, the differences between first and second screenings.

SupFig2A: The quality of TRF2 WB is low. Although 18 compounds were evaluated by WB, only 9 are depicted in the figure. It would be important to observe also the ones that up-regulate TRF2 for comparison, in particular because author conclude by stating they are potentially anti-aging agents by upregulating TRF2.

L249: Does this mean that the other seven compounds failed to decrease TRF2 in GRT-BJ-HELTRas cells?

Fig. 1C: DMSO bar is missing from graph

Fig S4A/B depicts the same data as in 3B/C (# and %). However, in S4B differences are between empty vector DMSO and TRF2 DMSO and AR, which is not consistent with 3B. The same for 3D/S4C, 3F/S4D.

L78: The sentence is quite long and could be rephrased for better reading

L121: Title for cloning strategy is incomplete

Author Response

In this work, Dr. Mai and colleagues aimed to identify compounds to counteract TRF2 oncogenic effects. The topic is quite interesting as TRF2 has been associated with poor outcomes and immunosuppression in several cancers.

Authors stemmed from their previous findings to devise a strategy for protein reduction instead of inhibition, thus avoiding DDR and pro-aging side effects, which is an interesting and original approach.

The model was BJ-HELTRas centered (immortalized oncogenic fibroblasts). Overall, the manuscript is well written and data well presented. Methods and controls were adequate.

However, despite authors previously shown that TRF2 expression can reverse tumorigenicity in mouse models through non-cell-autonomous effects, without inducing DDR (doi:10.1038/ncb2774,) a major concern is that although this study demonstrates a possible anti-oncogenic role of AR and AD by decreasing TRF2, the authors did not demonstrate that TRF2 reduction by AR and AD does not have DDR effects. Analysis of DDR proteins in GRT-BJ-HELTRas cels under AR and AD, for example, should be included.

We thank the reviewer for this comment and recognize that we did not demonstrate that TRF2 reduction induced by AR and AD does not have DDR effects. This comment has been taken into consideration and this limitation is further discussed in the revised version as follows “”Of note, we observe that AR and AD rescued the immunosuppressive and pro-angiogenic functions of TRF2 overexpression, two characteristics of TRF2 overexpression that we previously showed as DDR independent. Thus, we hypothesized that the AR and AD effects on tumor growth are DDR independent, a mechanism that remain to be fully described in further studies”. Indeed, our main objective was to determine whether the strategy to target TRF2 by chemicals is sufficient to decrease the pro-oncogenic activities of TRF2 overexpression dependently or independently of a DDR activation. We bring the proof of concept that hits from our drug screening significantly decrease in vivo the pro-tumorigenic effects of TRF2 overexpression essentially by the reactivation of immune system. Thus, we observe that the drugs inhibit the immune-suppressive and pro-angiogenic effects of TRF2, two oncogenic properties of TRF2 that we previously demonstrated to be DDR independent. Consequently, we hypothesized that the drugs counteract these pro-oncogenic effects independently of DDR activation. We agree with reviewer 1 that the characterization of drug effects on DDR should be explored. Since we now bring the proof of concept that pharmacologically targeting TRF2 emerge as a new therapeutical strategy, we plan to fully characterize the DDR dependent and DDR independent aspect of TRF2 targeting in a further study.

Additionally, there are some aspects that, if addressed, may improve the quality of the manuscript prior to consideration for publication.

L246, SupFig 1E: It is not clear what were, if any, the differences between first and second screenings.

The main difference between the first and the second screen is that the first was performed only with GRT-BJ-HELTRas cells, while the second included also non-transduced BJ-HELTRas cells or BJ-HELTRas transduced with an empty vector. During the second screening, GRT-BJ-HELTRas cells were treated with 10µM of the selected compounds or DMSO as performed in the primary screen. But in the second we added as controls non-transduced BJ-HELTRas cells or BJ-HELTRas cells transduced with an empty vector to discard false positives emitting red or green autofluorescence or affecting RFP or GFP proteins. We apologize for this lack of clarity. An explanation of the differences between the first and second screenings has been added in the result part and the description of the secondary screen has been modified in the legend of Figure S1.

SupFig2A: The quality of TRF2 WB is low. Although 18 compounds were evaluated by WB, only 9 are depicted in the figure. It would be important to observe also the ones that up-regulate TRF2 for comparison, in particular because author conclude by stating they are potentially anti-aging agents by upregulating TRF2.

We acknowledge that the quality of the western blots is not perfect but enough to a relative  estimation of TRF2 expression. We understand the remarks of the reviewer on the selection of the hits and we will clarify our strategy on drug selection. Our selection criteria were that hits must be defined as at least 20% of TRF2 dosage variation (either up or down) in at least 2 independent experiments. Using these criteria, from the 18 compounds evaluated by western blotting, 9 drugs reduced and one drug increased TRF2 protein levels in non-transduced BJ-HELTRas cells (as stated in Table S1). Among these drugs, we decided to choose two compounds for in vivo experiments to determine whether they could counteract the pro-oncogenic effects of TRF2 overexpression. To avoid the deleterious effects due to an extreme down-regulation of TRF2  in non-tumorigenic cells, we decided to avoid extreme conditions. Thus, we arbitrary chose to select the compounds that neither reduced to the highest nor to the lowest levels TRF2 dosage. Among them, AR and AD were corresponding to those criteria. While it could also be interesting to analyze the other compounds, only these two compounds were used in vivo to provide a proof of concept that a mild TRF2 targeting could mediate strong anti-tumorigenic potential. To clarify these results for the readers, Western blotting results encompassing the 18 compounds has been added in Figure S2. Moreover, further explanations of the choice of AR and AD for the in vivo experiments have been provided in the revised version of the manuscript as follows “We defined hits as compounds modulating by at least 20% of TRF2 dosage (either up or down). Using these criteria, from the 18 drugs evaluated by western blotting, 9 of them reduced and one increased endogenous TRF2 protein levels (Figure S2A-B, Table S1). We then decided to choose two compounds among these drugs for in vivo experiments to determine whether they could counteract the pro-oncogenic effects of TRF2 overexpression. To avoid DDR activation and side effects in non-tumorigenic cells, we selected compounds that neither reduced to the highest nor to the lowest levels TRF2 dosage. Among them, AR and AD were corresponding to those criteria.”.

L249: Does this mean that the other seven compounds failed to decrease TRF2 in GRT-BJ-HELTRas cells?

As explained above, only AR and AD were tested in vivo following our selection criteria. We agree with the reviewer that it could be interesting to test the seven other compounds in another study.

Fig. 1C: DMSO bar is missing from graph

The results reported in figure 1C represent RFP MFI/GFP MFI ratios from four different experiments where values were normalized by DMSO. Therefore, values of DMSO were 1 in each experiment.

Fig S4A/B depicts the same data as in 3B/C (# and %). However, in S4B differences are between empty vector DMSO and TRF2 DMSO and AR, which is not consistent with 3B. The same for 3D/S4C, 3F/S4D.

Figure 3 represents the cell number of each cell type among total living cells of each tumor. In contrast, figure S4 describe a percentage of these cell types among CD45+ immune cells meaning the different immune cell proportion. We observed that the number of CD45+ cells (global immune cell infiltration) was strongly increased by the drugs (Figure 3), while the global immune proportion of the different immune cell population are slightly modified in proportion among the immune cells (% of CD45+ cell; Figure S4). Our result show that TRF2 targeting strongly modify immune cell infiltration number but not immune cell proportion, ruling out this apparent discrepancy.

L78: The sentence is quite long and could be rephrased for better reading

We apologize for that. This sentence has been rephrased in the revised version.

L121: Title for cloning strategy is incomplete

The title for cloning strategy has been corrected.

Reviewer 2 Report

 In this manuscript, El Mai et al. investigate the effects of TRF2 modulation in the control of tumorigenesis and immunosuppression. To this end, they develop a screening strategy that utilizes fluorescently tagged cells to monitor the expression of TRF2 in the cells, and its variation with different FDA-approved drugs. Once they identify the potential hits, they narrow down their list to a couple of inhibitors with known mechanistic effects to examine how the inhibitors affect TRF2-induced tumor growth in-vivo. They conclude that the two drugs used--AR, and AD, restored immune cell filtration to counteract the TRF2-mediated tumor growth. While the study is of potential interest to cancer biology readers, I find several weaknesses in the current form that need addressing. 

One of the major concerns involves the selection of two compounds for further experiments and the lack of a clear rationale for choosing them. In the text, authors mention that these two compounds inhibited the expression of TRF2-RFP, while there were a few other compounds with effects higher than them, and still not chosen. My suspicion is that the other compounds didn't elicit much biological effects compared to these two, but it needs to be either shown, or discussed in the text. 

In Figure.2, authors examine the anti-tumorigenic effects of AR, and AD, specifically in the context of TRF2 over-expressing tumors. As presented, the figure seems to show a lot of variability even in similar conditions. It might be better to quantify the growth rates in all conditions examined since longitudinal data is available. To further prove their conclusion regarding the specificity of the drugs in TRF2-over-expressing context, I suspect a genetic knockdown validation might be necessary. Sure, these drugs decreased TRF2 expression, but they target different things, AR (GSK3B inhibitor), and AD (has mitochondrial effect), so my major concern is whether their effects are merely due to off-target effects, in addition to targeting TRF2 suppression. In that case, the interpretation of the results would be different. It is also worth noting that there are other compounds (mentioned earlier) with much higher TRF2 suppression that authors neither discussed, and examined. 

In Figure.3, authors examine the effects of AR, and AD, in modulating the immune response elicited by TRF2. Both AR, and AD enhance the recruitment of CD45+ immune cells, but in Figure.2B, that seems to be the case for empty-vector as well, especially for AD. Can authors comment on that? Also, authors need to show statistical comparison between Empty-vector, and TRF2-vector for DMSO control. Throughout Figure.3, authors compare the effects of AR, and AD, and statically compare their effects in TRF2-vector alone. However, when examined carefully, the empty-vector elicits almost identical effects in these drugs, can authors comment on the response, and how/why those responses in empty-vector are almost similar to TRF2-vector for the drug-treated groups? 

Beside immune-related response, I wonder whether AR and AD affect cell cycle progression in TRF2-vector tumors. If by extension, they prolong certain phases of cell-cycle, growth inhibition might be observed. Have authors examined their effects of cell cycle, and/or can you comment?

Some minor points in the manuscript: axis need to be shown for Figure 2B, 2C, if the conclusion is to show that GFP expression does not change. 

Overall, I find the manuscript interesting, and applaud the authors for following up on the screening results.

Author Response

 In this manuscript, El Mai et al. investigate the effects of TRF2 modulation in the control of tumorigenesis and immunosuppression. To this end, they develop a screening strategy that utilizes fluorescently tagged cells to monitor the expression of TRF2 in the cells, and its variation with different FDA-approved drugs. Once they identify the potential hits, they narrow down their list to a couple of inhibitors with known mechanistic effects to examine how the inhibitors affect TRF2-induced tumor growth in-vivo. They conclude that the two drugs used--AR, and AD, restored immune cell filtration to counteract the TRF2-mediated tumor growth. While the study is of potential interest to cancer biology readers, I find several weaknesses in the current form that need addressing. 

One of the major concerns involves the selection of two compounds for further experiments and the lack of a clear rationale for choosing them. In the text, authors mention that these two compounds inhibited the expression of TRF2-RFP, while there were a few other compounds with effects higher than them, and still not chosen. My suspicion is that the other compounds didn't elicit much biological effects compared to these two, but it needs to be either shown, or discussed in the text. 

We understand the concern of reviewer 2 and apologize for this lack of clarity. This is clarified in the revised version as foilows “We defined hits as compounds modulating by at least 20% of TRF2 dosage (either up or down). Using these criteria, from the 18 drugs evaluated by western blotting, 9 of them reduced and one increased endogenous TRF2 protein levels (Figure S2A-B, Table S1). We then decided to choose two compounds among these drugs for in vivo experiments to determine whether they could counteract the pro-oncogenic effects of TRF2 overexpression. To avoid DDR activation and side effects in non-tumorigenic cells, we selected compounds that neither reduced to the highest nor to the lowest levels TRF2 dosage. Among them, AR and AD were corresponding to those criteria.. Indeed, our selection criteria were that hits must be defined as at least 20% of TRF2 dosage variation (either up or down) in at least 2 independent experiments. Using these criteria, from the 18 compounds evaluated by western blotting, 9 drugs reduced and one drug increased TRF2 protein levels in non-transduced BJ-HELTRas cells (as stated in Table S1). Among these drugs, we decided to choose two compounds for in vivo experiments to determine whether they could counteract the pro-oncogenic effects of TRF2 overexpression. To avoid deleterious effects in non-tumorigenic cells, we decided to avoid extreme conditions. Thus, we arbitrary chose to select the compounds that neither reduced to the highest nor to the lowest levels TRF2 dosage. Among them, AR and AD were corresponding to those criteria. While it could also be interesting to analyze the other compounds, only these two compounds were used in vivo to provide a proof of concept that a mild TRF2 targeting could mediate strong anti-tumorigenic potential. To clarify these results for the readers, Western blotting results encompassing the 18 compounds has been added in Figure S2. Moreover, further explanations of the choice of AR and AD for the in vivo experiments have been provided in the revised version of the manuscript.

In Figure.2, authors examine the anti-tumorigenic effects of AR, and AD, specifically in the context of TRF2 over-expressing tumors. As presented, the figure seems to show a lot of variability even in similar conditions. It might be better to quantify the growth rates in all conditions examined since longitudinal data is available.

We would like to thank reviewer 2 for this suggestion of quantifying growth rate. We calculated the growth rate of each condition by considering the tumor volume before treatment as 100% (see figure below). We analyzed the growth in kinetics since day 15 meaning the beginning of the treatment until the end. In those graphs, we clearly observe that AR and AD significantly dampened tumor growth only of TRF2-overexpressing tumors while no differences are seen in the empty condition between treated and u  ntreated tumors. These graphs have been added to the revised manuscript in the New Figure 2.

To further prove their conclusion regarding the specificity of the drugs in TRF2-over-expressing context, I suspect a genetic knockdown validation might be necessary. Sure, these drugs decreased TRF2 expression, but they target different things, AR (GSK3B inhibitor), and AD (has mitochondrial effect), so my major concern is whether their effects are merely due to off-target effects, in addition to targeting TRF2 suppression. In that case, the interpretation of the results would be different. It is also worth noting that there are other compounds (mentioned earlier) with much higher TRF2 suppression that authors neither discussed, and examined. 

A TRF2 knock-down using ShRNA strongly reduces tumor intake and tumor growth, as we and others reported previously (for instance Biroccio et al, NCB 2013). In the experiments described in the present manuscript, drug treatment was performed after tumor establishment (day 15). Therefore, adding drug treatments on TRF2 compromised cancer cells presenting already  a tumor growth defect would certainly not lead to detectable effect. On the contrary, we observed in the empty control condition, meaning cancer cells with only endogenous TRF2 expression, that drug treatment does not alter tumor growth (Figure 2). We acknowledge that considering their already described functions, AR and AD might inhibit tumor growth independently of TRF2 suppression. However, as described in Figure 2 and 3, AD and AR inhibit tumor growth and enhance immune response of TRF2 overexpressing tumors but not of tumors transduced with the empty vector. These results indicate that the effects of these two compounds on tumor progression rely mainly on their ability to inhibit TRF2 and not really on TRF2 independent effects of the drugs. 

In Figure.3, authors examine the effects of AR, and AD, in modulating the immune response elicited by TRF2. Both AR, and AD enhance the recruitment of CD45+ immune cells, but in Figure.2B, that seems to be the case for empty-vector as well, especially for AD. Can authors comment on that?

We thank reviewer 2 for this comment and recognize that the effects of AD on CD45+ cells in the empty condition have not been discussed. Further investigations would be required to understand how AD promotes the recruitment of CD45+ cells. We previously described that TRF2 promotes tumorigenesis by inhibiting the recruitment and activation of NK cells. Even though AD acts on CD45+ cells in the empty vector condition, this compound significantly enhances NK cell recruitment and activation only in TRF2-overexpressing tumors and not in the empty vector condition compared to DMSO treated mice. This indicate that AD counteracts TRF2 dependent immune suppressive effects despite potential other effects on CD45+ cell recruitment.

Also, authors need to show statistical comparison between Empty-vector, and TRF2-vector for DMSO control. Throughout Figure.3, authors compare the effects of AR, and AD, and statically compare their effects in TRF2-vector alone. However, when examined carefully, the empty-vector elicits almost identical effects in these drugs, can authors comment on the response, and how/why those responses in empty-vector are almost similar to TRF2-vector for the drug-treated groups? 

Throughout Figure 3, statistical analyses have always been performed to compare treated or untreated tumors in either empty vector or TRF2 tumors. For more clarity, we decided to show statistical comparisons only when significantly different. Therefore, the differences seen after treatment in empty-vector are not significant (except when stated in the figure). As shown in figure S3, AR and AD treatment reduces TRF2 levels in the empty vector condition. We previously characterized the in vivo anti-tumoral effects of TRF2 knockdown using ShRNA. Therefore, tendencies toward similar effects between the empty-vector condition and TRF2 vector in the current manuscript might be explained by the reduced TRF2 levels in the empty condition.

Beside immune-related response, I wonder whether AR and AD affect cell cycle progression in TRF2-vector tumors. If by extension, they prolong certain phases of cell-cycle, growth inhibition might be observed. Have authors examined their effects of cell cycle, and/or can you comment?

Contrary to non-tumorigenic cells, we previously described that TRF2 knock down in tumor cells does not impact neither cell cycle nor viability of these cells in vitro. However, knock down of TRF2 strongly reduced the tumorigenicity in vivo by an extrinsic mechanism depending on Natural killer cells inhibition. For these reasons, we did not analyze the effects of AR and AD on cell cycle in TRF2 vector tumors.

Some minor points in the manuscript: axis need to be shown for Figure 2B, 2C, if the conclusion is to show that GFP expression does not change. 

We apologize for this omission. Axis has been added in the new figure version.

Overall, I find the manuscript interesting, and applaud the authors for following up on the screening results.

We thank reviewer 2 for his interest on the manuscript.

Reviewer 3 Report

The authors developed a cell based system to follow the stability of ectopically expressed shelterin complex protein TRF2 for screening of novel anti-TRF2 anticancer compounds. Using this system they screened library of ~400 FDA-approved compounds and identified two drugs Alexidine 2HCl and AR-A014418 destabilizing TRF2. To prove their anticancer efficacy, they used immunocompromised NMRI mouse xenografted with human preputium-derived fibroblasts BJ transformed with SV40/telomerase and oncogenic Ras (BJ-HELTRas) with TRF2 overexpression. Both compounds suppressed TRF2-overexpressing BJ-HELTRas-derived tumor growth. The effect of both drugs on TRF2-modulated tumor microenvironment was examined as well. Both drugs suppressed TRF2-enhanced recruitment of immunosuppressive cells into tumor (Matrigel plug-containing un/treated BJ-HELTRas cells) and blocked TRF2-induced angiogenesis. Thus both drugs have potential to be used in clinics to treat TRF2-overexpressing cancers.

In conclusion, it is very interesting and – at most – technically well done study definitely of interests of cancer research community. The formal elaboration of the manuscript is of a high standard.

Below I have some questions and suggestions to improve the study before publication in Cancers journal.

General comments:

A limitation of the study is that it is based on just one in vivo (rather artificial) tumor model (using immunocompromised animals) and no targets of both compounds have been identified. However, the value of the study can be increased by mechanistic explanation of the effects of both drugs on regulation of TRF2 levels, which is unclear, and can be relatively easily complemented.

Specific comments:

Line 253 - 255: The effect of both compounds on mRNA levels driven both by endogenous and ectopic promoter (in TRF2-overexpressing BJ-HELTRas cells) is confusing without mechanistic explanation. If I understand correctly, the screening strategy is based on the effect on TRF2 protein stability driven by ectopic promoter and thus not on its transcription regulation. Authors show (Figure S3C) that both drug suppress endogenous TRF2 transcript level. This is somehow explained in discussion as that 'AR and AD target multiple levels of TRF2 regulation'. However, to show unequivocally the effect of both drug on TRF2 protein/mRNA stability, supporting experiments are needed. I would suggest to use cycloheximide to inhibit translation and to follow TRF2 levels (+- drugs) by western blotting. Alternatively, the effect of both drugs on TRF2 mRNA stability should be proved and mechanism explained.

Line 161: Please, include sex of animals (males?).

Line:169: Please, include the concentration/volume of DMSO applied, application way and volume of both drugs.

Line 175: Please, include the concentration of DMSO.

Line 318: Please, specify the antigens used for discriminating MDSC cells (CD11b+ GR1+) in Figure 3 legend/Methods section).

The western blots are of very poor quality overall.

Minor issues (some typos, etc.):

Line 51: Please, use lower case for Dyskeratosis congenita

Line 81: 'inhibit' should be 'inhibits'

Line 84: 'IFNg' should be 'IFNgamma'

Line 109 and further: Please, use the term 'western blotting' for technique (not 'blot')

Line 126: 'florescent'

Line 128: 'picornaviridae' (better should be 'Picornaviridae')

Line 412: '-actin'

Figure 3: Commonly used abbreviation 'No.' instead of "Nbr' in y-axis legends would be more appropriate

Figure S1: Please, use dots for decimal points instead of commas

Figure S2: 'int.ensity'

Figure S3: 'measurment'

Table 1: The legend should be improved (uM, etc.).

US English should be uniformed throughout the main text and figures (e.g., analysed vs. analyzed, etc.).

Capitals for drugs (such as Gemcitabine, etc.), should be avoided.

Careful editing of the figure legends is needed.

Author Response

The authors developed a cell based system to follow the stability of ectopically expressed shelterin complex protein TRF2 for screening of novel anti-TRF2 anticancer compounds. Using this system they screened library of ~400 FDA-approved compounds and identified two drugs Alexidine 2HCl and AR-A014418 destabilizing TRF2. To prove their anticancer efficacy, they used immunocompromised NMRI mouse xenografted with human preputium-derived fibroblasts BJ transformed with SV40/telomerase and oncogenic Ras (BJ-HELTRas) with TRF2 overexpression. Both compounds suppressed TRF2-overexpressing BJ-HELTRas-derived tumor growth. The effect of both drugs on TRF2-modulated tumor microenvironment was examined as well. Both drugs suppressed TRF2-enhanced recruitment of immunosuppressive cells into tumor (Matrigel plug-containing un/treated BJ-HELTRas cells) and blocked TRF2-induced angiogenesis. Thus both drugs have potential to be used in clinics to treat TRF2-overexpressing cancers.

In conclusion, it is very interesting and – at most – technically well done study definitely of interests of cancer research community. The formal elaboration of the manuscript is of a high standard.

Below I have some questions and suggestions to improve the study before publication in Cancers journal.

General comments:

A limitation of the study is that it is based on just one in vivo (rather artificial) tumor model (using immunocompromised animals) and no targets of both compounds have been identified. However, the value of the study can be increased by mechanistic explanation of the effects of both drugs on regulation of TRF2 levels, which is unclear, and can be relatively easily complemented.

 We appreciate reviewer’s interest in understanding the mechanism behind the regulation of TRF2 by AR and AD. As discussed in the manuscript, the mechanisms of TRF2 regulation by these compounds remain to be determined. In fact, the results of this manuscript have initiated projects investigating in a more global view how the Wnt pathway and mitochondrial function could modulate TRF2 protein levels. These projects will therefore provide further insights on the regulation of TRF2 by AR and AD.

Specific comments:

Line 253 - 255: The effect of both compounds on mRNA levels driven both by endogenous and ectopic promoter (in TRF2-overexpressing BJ-HELTRas cells) is confusing without mechanistic explanation. If I understand correctly, the screening strategy is based on the effect on TRF2 protein stability driven by ectopic promoter and thus not on its transcription regulation. Authors show (Figure S3C) that both drug suppress endogenous TRF2 transcript level. This is somehow explained in discussion as that 'AR and AD target multiple levels of TRF2 regulation'. However, to show unequivocally the effect of both drug on TRF2 protein/mRNA stability, supporting experiments are needed. I would suggest to use cycloheximide to inhibit translation and to follow TRF2 levels (+- drugs) by western blotting. Alternatively, the effect of both drugs on TRF2 mRNA stability should be proved and mechanism explained.

We recognize that we have not described the mechanism by which AR and AD regulates TRF2 levels. Using an ectopic promoter (SFFV promoter), the screening strategy that allowed us to select AR and AD was indeed following TRF2 exogenous protein levels. Western blotting experiments and RT-qPCR data described that both protein and mRNA levels of TRF2 were affected by both drugs. As discussed in the manuscript, mechanistic insights of these regulations remain to be determined. In this manuscript, we bring a proof of concept that pharmacologically targeting TRF2 emerge as a new therapeutic anti-cancer strategy. We plan now to further investigate the mechanism by which both drugs and more globally the Wnt pathway and mitochondrial dysfunction modulate TRF2.

Line 161: Please, include sex of animals (males?).

 Sex of animals (female) have been included in the revised version of the manuscript

Line:169: Please, include the concentration/volume of DMSO applied, application way and volume of both drugs.

Information concerning the concentration of DMSO applied (45%), application way (intra-peritoneal), and volume of drugs/DMSO (100µL) have been included in the revised version of the manuscript.

 Line 175: Please, include the concentration of DMSO.

 Concentration of DMSO (1%) used have been added to the manuscript.

Line 318: Please, specify the antigens used for discriminating MDSC cells (CD11b+ GR1+) in Figure 3 legend/Methods section).

We apologize for omitting to specify the antigens used for discriminating MDSC cells. This specification has been added in Figure 3 legend of the revised manuscript.

The western blots are of very poor quality overall.

 We acknowledge that the quality of the western blots could be improved and apologize for that.

Minor issues (some typos, etc.):

Line 51: Please, use lower case for Dyskeratosis congenita

Line 81: 'inhibit' should be 'inhibits'

Line 84: 'IFNg' should be 'IFNgamma'

Line 109 and further: Please, use the term 'western blotting' for technique (not 'blot')

Line 126: 'florescent'

Line 128: 'picornaviridae' (better should be 'Picornaviridae')

Line 412: '-actin'

Figure 3: Commonly used abbreviation 'No.' instead of "Nbr' in y-axis legends would be more appropriate

Figure S1: Please, use dots for decimal points instead of commas

Figure S2: 'int.ensity'

Figure S3: 'measurment'

Table 1: The legend should be improved (uM, etc.).

US English should be uniformed throughout the main text and figures (e.g., analysed vs. analyzed, etc.).

Capitals for drugs (such as Gemcitabine, etc.), should be avoided.

Careful editing of the figure legends is needed.

We thank reviewer 3 for underlying these minor issues and apologies for these mistakes. Appropriate modifications have been performed in the revised version of the manuscript.

Round 2

Reviewer 1 Report

In the revised version the authors clarified all my questions. This is an improved version.

Author Response

We kindly thank reviewer 2 for his interest on the manuscript, for his comments and for providing us the opportunity to improve its quality.

Reviewer 3 Report

Authors significantly improved the manuscript. 

However, several typos still left omitted in the revised version, especially in figures. It seems sort of the old version of figures was resubmitted. 

I have additional comments:

Line 114: Please, do not use designation "Laemmli buffer" for buffer of  the composition mentioned in "SDS-PAGE and Western blotting section", i.e., 500 mM Tris-HCl, 100 mM DTT, 2% SDS, 0.1% bromophenol blue, 10% glycerol, pH 6.8. Please, use instead "SDS sample lysis buffer" or similar.

Laemmli buffer is defined as 62.5 Tris-HCl (pH 6.8), 2% SDS, 10% glycerol, 5% 2-mercaptoethanol, and 0.001% bromophenol blue, see Laemmli, U. Cleavage of Structural Proteins during the Assembly of the Head of Bacteriophage T4. Nature 227, 680–685 (1970). https://doi.org/10.1038/227680a0.

Using 500 mM Tris instead of correct 62.5 mM in sample buffer might explain non-optimal quality of blots.

In fact, the description of the sample preparation for SDS-PAGE is quite confusing and should be precisely stated.

Author Response

We would like to thank reviewer 3 for its careful editing comments of the manuscript and we apologize for the mistakes that were remaining. Typos that were not corrected in Figure 3 and Figure S1 have been now modified hoping that no other mistake is left uncorrected. Moreover, a thorough modification of the "SDS-PAGE and Western blotting section" has been performed.